# Effects of Cathode GDL Gradient Porosity Distribution along the Flow Channel Direction on Gas–Liquid Transport and Performance of PEMFC

**DOI:** 10.3390/polym15071629

**Published:** 2023-03-24

**Authors:** Ruijie Zhu, Zhigang Zhan, Heng Zhang, Qing Du, Xiaosong Chen, Xin Xiang, Xiaofei Wen, Mu Pan

**Affiliations:** 1State Key Laboratory of Advanced Technology for Materials Synthesis and Processing, Wuhan University of Technology, Wuhan 430070, China; 2Donghai Laboratory, Zhoushan 316022, China; 3Hubei Key Laboratory of Fuel Cells, Wuhan 430070, China; 4Foshan Xianhu Laboratory of the Advanced Energy Science and Technology Guangdong Laboratory, Xianhu Hydrogen Valley, Foshan 528200, China

**Keywords:** PEMFC, GDL, gradient porosity, gas–liquid transport, multi-physical distribution

## Abstract

The gas diffusion layer (GDL) is an important component of proton exchange membrane fuel cells (PEMFCs), and its porosity distribution has considerable effects on the transport properties and durability of PEMFCs. A 3-D two-phase flow computation fluid dynamics model was developed in this study, to numerically investigate the effects of three different porosity distributions in a cathode GDL: gradient-increasing (Case 1), gradient-decreasing (Case 3), and uniform constant (Case 2), on the gas–liquid transport and performance of PEMFCs; the novelty lies in the porosity gradient being along the channel direction, and the physical properties of the GDL related to porosity were modified accordingly. The results showed that at a high current density (2400 mA·cm^−2^), the GDL of Case 1 had a gas velocity of up to 0.5 cm·s^−1^ along the channel direction. The liquid water in the membrane electrode assembly could be easily removed because of the larger gas velocity and capillary pressure, resulting in a higher oxygen concentration in the GDL and the catalyst layer. Therefore, the cell performance increased. The voltage in Case 1 increased by 8% and 71% compared to Cases 2 and 3, respectively. In addition, this could ameliorate the distribution uniformity of the dissolved water and the current density in the membrane along the channel direction, which was beneficial for the durability of the PEMFC. The distribution of the GDL porosity at lower current densities had a less significant effect on the cell performance. The findings of this study may provide significant guidance for the design and optimization of the GDL in PEMFCs.

## 1. Introduction

Currently, reducing environmental pollution and achieving clean and efficient energy utilization are imperative global issues. A proton exchange membrane fuel cell (PEMFC) can convert the chemical energy stored in reactants into electrical energy. It has many advantages, such as high energy conversion efficiency and power density, rapid start-up, and environmental friendliness. Therefore, it is regarded as one of the most promising new energy conversion devices to replace traditional energy sources [1,2,3].

As the operating temperature of a PEMFC is lower than other types of fuel cell, the water generated in the catalytic layer (CL) may exist in liquid form. Water is essential for PEMFCs, because protons can only efficiently cross the membrane when it is sufficiently hydrated [4]. However, there is also has a negative impact on cell performance when the liquid water in the membrane electrode assembly (MEA) is excessive [5]. As this blocks the gas transport path and reduces the effective gas diffusivity, causing a higher transport resistance and thus performance degradation of PEMFCs. Therefore, to realize stable operation and better performance of PEMFCs, it is important to have an in-depth understanding of the gas–liquid transport in the MEA.

Many in-depth studies have been conducted to improve the efficiency of reactant transport and water management ability, by optimizing the structure of key components and the operating conditions of PEMFCs, including the effects of channel size and shape [6,7], the flow field distribution zone structure [8], and operating conditions, including the temperature, back pressure, gas humidification, and stoichiometric ratio of reactants [9,10,11]. The transport of liquid water in a MEA is determined by the capillary pressure and material properties [12]. The gas diffusion layer (GDL) and microporous layer (MPL) are collectively referred to as gas diffusion media (GDM) [13], whose pore structure is key to the gas–liquid transport in MEA. Many studies have been carried out to optimize the structure of the GDL to achieve an efficient gas–liquid transport. Shi et al. [14] developed a 2-D PEMFC model and studied the effect of cracks in the MPL on PEMFC performance. The results indicated that liquid water in the CL can be expelled into the gas channel through these cracks, resulting in a lower gas-transport resistance, and thus avoiding flooding. Zhang et al. [15] compared and analyzed the porosity distribution of different GDLs using the stochastic reconstruction method and determined anisotropic effective transport properties using a pore-scale model. A 3-D model was established by Jha et al., to study the effect of GDL porosity on the performance of a high-temperature PEMFC [16]. They found that the cell performance could be improved when the GDL porosity was in the range of 0.5–0.6. Xia et al. [17] developed a 3-D PEMFC model and found that increasing the porosity and thickness of the GDL increased the oxygen concentration under the rib, consequently improving the uniformity of oxygen distribution in the GDL. Sim et al. [18] experimentally investigated the effect of the polytetrafluoroethylene (PTFE) content in the GDL on PEMFC performance. They found that the performance deteriorated as the PTFE content in the substrate increased.

Moreover, the gradient distribution of porosity, PTFE, etc., in the through-plane direction of porous materials in a MEA has a significant impact on PEMFC performance, hence attracting much attention. Lim et al. [19] combined two GDLs with different PTFE contents to form a complete GDL. They found that a PEMFC with an untreated bottom layer and a PTFE-treated top layer of GDL exhibited optimal performance under low humidity conditions. Ko et al. [20] studied the effect of the porosity gradient in the through-plane direction of GDL on cell performance. The results showed that the cell with the medium-gradient GDL exhibited the best performance under low-humidity conditions. Lim et al. [21] developed a numerical model of a PEMFC with a serpentine flow channel and found that the current density distribution of the PEMFC was more uniform when the GDL had a porosity gradient in the through-plane direction on both the inlet and outlet sides. Zhan et al. [22,23] numerically analyzed the effect of porosity gradient in the through-plane direction of the GDL on the liquid water flux and gas diffusion flux. A GDL with decreasing porosity was more conducive to removing liquid water, which could effectively alleviate flooding. Shangguan et al. used the volume of fluid (VOF) method to numerically study the effect of GDL porosity and contact angle on water distribution [24]. It was found that a GDL with a “V” shaped porosity distribution in the through-plane direction achieved rapid water removal from hydrophobic surfaces. Habiballahi et al. and Carcadea et al. [25,26] developed numerical models of the PEMFC and found that when the GDL porosity decreased in the direction from the channel to the CL, liquid at a high current density could be effectively reduced, consequently improving the water management of the PEMFCs. All the above studies mainly focused on the design of porosity, contact angle, etc. in the through-plane direction.

In fact, a non-uniform porosity distribution of GDL in the in-plane direction also has a significant effect on gas–liquid transport and cell performance, and there have been a few studies in recent years analyzing the effects of a porosity gradient distribution in the GDL along the channel direction. Kanchan et al. [27,28] investigated the effect of GDL porosity variation along the channel direction on the cell performance, and the cell was operated at high temperature with a single-phase flow. They concluded that the GDL with a logarithmically decreasing porosity gradient improved the cell performance. Yang et al. [29] numerically analyzed the effects of GDL porosity variation with various increasing gradients from the gas inlet to the outlet, they concluded that a cell with an increasing porosity gradient could increase the current density. Sim et al. [30] achieved a non-uniform porosity distribution by changing the compression ratio of the GDL, the outlet area was less compressed than the inlet area and hence had a larger porosity, and the cell performance increased.

These few studies, however, did not consider the correction of relevant parameters due to the change of porosity, such as the liquid water permeability, effective gas diffusion coefficient, etc. The mechanism and effect of the porosity gradient distribution in a GDL along the channel direction were not clearly explained, due to the complexity of multi-physical field coupling; and because of the varied operating conditions, some of the conclusions drawn are inconsistent with each other. Therefore, a 3-D two-phase model was built herein, to compare and analyze the effects of GDL gradient porosity distributions, including an increasing gradient, decreasing gradient, and uniform porosity along the channel direction, on the gas–liquid transport and cell performance. Transport parameters such as permeability and gas diffusion coefficients were corrected accordingly. This article is organized as follows: Section 2 gives model descriptions, including a geometric model, model assumption, governing equation, algorithm, boundary conditions, and model validation; Section 3 provides the results and discussion; Section 4 gives the conclusions obtained, which can be used as a reference for the optimal design of the GDL of PEMFCs.

## 2. Model Descriptions

### 2.1. Geometric Model

In this study, a 3-D, multi-component, two-phase PEMFC model was developed. The geometric domain is illustrated in Figure 1. It comprises two current collectors, flow channels, GDLs, MPLs, CLs, and a proton exchange membrane (PEM). The geometric parameters and porosities of each component of the model are listed in Table 1.

### 2.2. Model Assumptions

To simplify the complexity of the calculation and improve the efficiency, the following reasonable assumptions were made for this model:(1)The system works at a steady state.(2)The reaction gas is considered an incompressible ideal gas.(3)The reactant is set to laminar flow, owing to the low gas-flow velocity and Reynolds number (The maximum Re in this study is 47, much less than 2000) [31].(4)Ionomer water is generated by the electrochemical reaction in the CL [32,33].(5)The temperatures of the walls and inlet channels are constant.(6)All the components in the PEMFC are isotropic except for the cathode GDL.(7)Contact resistance between components is ignored [34].(8)The effect of gravity is ignored [34].

### 2.3. Governing Equations

The basic governing equations have been widely used [35,36,37,38,39,40,41,42], but for clarity, they are presented in this section, including equations of mass conservation, momentum conservation, energy conservation, electrochemical reactions, dissolved water transport, and liquid water transport.

#### 2.3.1. The Conservation of Mass

The mass conservation equation in this model is described as:(1)$$\nabla \cdot \left(\rho \varepsilon \vec{u}\right)=S_{m}$$
where *ρ* is the density, *ε* is the volume fraction of open pores, which can be expressed as $\varepsilon =\varepsilon_{0}\left(1-s\right)$ [43]; $\varepsilon_{0}$ is the intrinsic porosity of the porous layer; and *s* is the liquid saturation, $\vec{u}$ is the fluid velocity, and $S_{m}$ is the mess source term.

#### 2.3.2. The Conservation of Momentum

The momenturm conservation equation in this model is described as:(2)$$\nabla \cdot \left(\varepsilon \rho \vec{u}\vec{u}\right)=-\varepsilon \nabla p+\nabla \cdot \left(\varepsilon \mu \nabla \vec{u}\right)+S_{u}$$
where ρ is the density, p is the pressure, μ is the viscosity, and $S_{u}$ is the momentum source term.

#### 2.3.3. The Conservation of Energy

The energy conservation equation in this model is described as:(3)$$\nabla \cdot \left(\varepsilon \rho c_{p}\vec{u}T\right)=\nabla \cdot \left(k^{eff}\nabla T\right)+S_{T}$$
where $c_{p}$ is the constant-pressure specific heat, T is the temperature, $k^{eff}$ is the effective heat conductivity, and $S_{T}$ is the heat source term.

#### 2.3.4. The Conservation of Species

The species conservation equation in this model is described as:(4)$$\nabla \cdot \left(\varepsilon \vec{u}c_{i}\right)=\nabla \cdot \left(D_{i}^{eff}\nabla c_{i}\right)+S_{i}$$
where $c_{i}$ is the concentration of the *i*th species, and $D_{i}^{eff}$ is the effective diffusion coefficient

The volumetric source terms (kg·m^−3^·s^−1^) for $H_{2}$, $O_{2}$, and the water in the CL due to electrochemical reactions are solved as follows:(5)$$S_{H_{2}}=-\frac{M_{w,H_{2}}}{2F}j_{an}$$
(6)$$S_{O_{2}}=-\frac{M_{w,O_{2}}}{4F}j_{ca}$$
(7)$$S_{H_{2}O}=\frac{M_{w,H_{2}O}}{2F}j_{ca}$$
where $M_{w,H_{2}O}$, $M_{w,O_{2}}$, and $M_{w,H_{2}}$ are the molecular masses of water, oxygen, and hydrogen, respectively; *F* is the Faraday constant; and $j_{an}$ and $j_{ca}$ are the source terms of current densities on the anode and cathode sides, which can be described using the Butler–Volmer equations:(8)$$j_{an}=A_{an}j_{0,a}^{ref}{\left(\frac{C_{H_{2}}}{C_{H_{2}}^{ref}}\right)}^{\gamma_{an}}\left(e^{\frac{\alpha_{an}F\eta_{an}}{RT}}-e^{\frac{-\alpha_{an}F\eta_{an}}{RT}}\right)$$
(9)$$j_{ca}=A_{ca}j_{0,c}^{ref}{\left(\frac{C_{O_{2}}}{C_{O_{2}}^{ref}}\right)}^{\gamma_{ca}}\left(-e^{\frac{\alpha_{ca}F\eta_{ca}}{RT}}+e^{\frac{-\alpha_{ca}F\eta_{ca}}{RT}}\right)$$
where *A* is the specific active surface area, because of the liquid coverage, and is modeled as: $A=\left(1-s\right)A_{0}$; $j_{0}^{ref}$ is the reference exchange current density; $C^{ref}$ is the local species reference concentration; *γ* is the exponent of concentration; R  is the universal gas constant; *α* is the transfer coefficient; “*an*” and “*ca*” represent the anode and cathode; *η* is the overpotential, which in the anode is $\eta =\varphi_{ele}-\varphi_{ion}$ and in the cathode is $\eta =\varphi_{ele}-\varphi_{ion}-U_{0}$; and $U_{0}$ is the open-circuit potential:(10)$$U_{0}=1.23-0.0009\left(T-298\right)+0.575\left(\mathrm{RT}/F\right)\left(\log\;\;p_{H_{2}}^{2}p_{O_{2}}\right)$$

#### 2.3.5. The Electric and Proton Transport

The electric and proton transport in both the anode and cathode CLs is described as follows:

Proton transport:(11)$$\nabla \cdot \left(k_{ion}^{eff}\nabla \varphi_{ion}\right)+S_{ion}=0$$

Electric transport:(12)$$\nabla \cdot \left(k_{ele}^{eff}\nabla \varphi_{ele}\right)+S_{ele}=0$$

In the cathode CL, $S_{ion}$=$-j_{ca}$ and $S_{ele}=j_{ca}$; and in the anode CL, $S_{ele}$=$-j_{an}$, $S_{ion}=j_{an}$, $k_{ele}^{eff}$ is the effective electron conduction rate: $k_{ele}^{eff}=k_{s}\left(1-s-\varepsilon \right)$, $k_{ion}^{eff}$ is the effective proton conduction rate: $k_{ion}^{eff}=k_{m}\omega^{1.5}$, ω is the volume fraction of the ionomer in the CL, $\varphi_{ion}$ is the proton potential, and $\varphi_{ele}$ is the electron potential.

#### 2.3.6. The Dissolved Water Transports

Dissolved water exists in the CLs (ionomers) and the membrane, whose generation and transportation are described in [44]:(13)$$\nabla \cdot \left({\vec{l}}_{m}^{\;}\frac{n_{d}}{f}M_{w,H_{2}O}\right)=\nabla \cdot \left(M_{w}D_{w}^{i}\;\nabla \lambda \right)+S_{\lambda }+S_{gd}+S_{ld}$$
where $M_{w,H_{2}O}$ is the molecular masses of water, ${\vec{l}}_{m}^{\;}$ is the ionic current density calculated as: ${\vec{l}}_{m}^{\;}=-\sigma_{mem}\nabla \varphi_{mem}$, $\varphi_{mem}$ is the ionic phase potential, $n_{d}$ is the osmotic drag coefficient, $D_{w}^{i}$ is the diffusion coefficient of water content, λ is the water content, $S_{\lambda }$ is the water generation rate in the catalyst layer, $S_{gd}$ is the mass change rate between gas and dissolved phases, and $S_{ld}$ is the mass change rate between liquid and dissolved phases.

The $S_{gd}$ and $S_{ld}$ are described as in [44,45]:(14)$$S_{gd}=\left(1-s\right)\gamma_{gd}M_{w}\frac{\rho_{i}}{EW}\left(\lambda_{eq}-\lambda \right)$$
(15)$$S_{ld}=s\gamma_{ld}M_{w}\frac{\rho_{i}}{EW}\left(\lambda_{eq}-\lambda \right)$$
where $\gamma_{gd}$ and $\gamma_{ld}$ are the mass exchange rate constants between the gas and liquid, *EW* is the equivalent weight of the membrane, $\rho_{i}$ is the membrane density, and $\lambda_{eq}$ is the equilibrium water content, which was computed as in [46]:(16)$$\lambda_{eq}=\{\begin{array}{cc} 0.043+17.81a-39.85a^{2}+36.0a^{3} & 0\leq a\leq 1\\ 14.0+1.4\left(a-1\right) & 1\leq a\leq 3 \end{array}$$
where a is the water activity defined as: $a=p_{wv}/p_{sat}$, $p_{wv}\;$ is the water vapor partial pressure, and $p_{sat}$ is the saturation pressure.

#### 2.3.7. The Liquid Water Transport

The liquid water in the cathode and anode CLs, MPLs, and GDLs is driven by capillary pressure.
(17)$$\nabla \cdot \left(\frac{\rho_{l}KK_{r}}{\mu_{l}}\nabla (p_{c}+p_{g})\right)+S_{gl}-S_{ld}=0$$
(18)$$P_{c}=P_{l}-P_{g}=\frac{-\sigma \cos\theta }{{\left(K/\varepsilon_{0}\right)}^{0.5}}\left(1.417s-2.12s^{2}+1.263s^{3}\right)$$
where $K,\;K_{r},\;\mu_{l},\rho_{l},\;p_{c},p_{g},\;\;S_{gl}$ represent the absolute permeability, relative permeability, liquid dynamic viscosity, liquid density, capillary pressure, gas pressure, and the mass change rate between the gas and liquid water, respectively.

$S_{gl}$ are determined using the following expressions [34,44]:(19)$$S_{gl}=\{\begin{array}{cc} \gamma_{e}\varepsilon_{0}s\frac{p_{wv}-p_{sat}}{RT}, & p_{wv}\leq p_{sat}\\ \gamma_{c}\varepsilon_{0}\left(1-s\right)\frac{p_{wv}-p_{sat}}{RT}, & p_{wv}>p_{sat} \end{array}$$
where $\gamma_{e}$ is the evaporation rate coefficient and $\gamma_{c}$ is the condensation rate coefficient, which usually takes the value of 1.3 (s^−1^) [44], $p_{wv}$ is the water vapor pressure, and $p_{sat}$ is the saturated vapor pressure.

Moreover, as parameters such as the permeability and gas diffusivity of GDL vary with the porosity, the relevant parameters are corrected as follows [41]:(20)$$K=\frac{\varepsilon_{0}}{8{\left(\varepsilon_{0}\right)}^{2}}\frac{{\left(\varepsilon_{0}-\varepsilon_{p}\right)}^{\left(\alpha +2\right)}d_{f}^{2}}{{\left(1-\varepsilon_{p}\right)}^{\alpha }{\left[\left(\alpha +1\right)\varepsilon_{0}-\varepsilon_{p}\right]}^{2}}$$
where α and $\varepsilon_{p}$ are constants whose values depend on the fiber arrangement and flow direction relative to the fiber plane (in the 3D model, α is taken as 0.661 and $\varepsilon_{p}$ is 0.037), and $d_{f}$ is the fiber diameter.

The gas phase species diffusivity is expressed as follows [47]:(21)$$D_{i}^{eff}=D_{i}\varepsilon_{0}^{1.5}{\left(1-s\right)}^{1.5}$$
where $D_{i}^{0}$ is the mass diffusivity at reference temperature $T_{0}$ and pressure $p_{0}$.

### 2.4. Numerical Methodology

The model in this study was solved using computational fluid dynamics (CFD) software (ANSYS Fluent 2022 R1). Based on the finite volume method, the above control equations were solved with double precision. The SIMPLE algorithm was used to deal with the velocity-pressure coupling in the momentum equation. The first-order headwind method was used as the interpolation function. In order to ensure convergence stability, an appropriate relaxation factor was applied to each variable. The convergence iteration standard of the energy equation was 10^−6^, and the other equations used 10^−3^. The porosity gradient distribution in GDL was realized using the UDF function, and the relevant parameters were also modified.

### 2.5. GDL Gradient Porosity Calculation Case Study

The effects of three different gradient porosities in the cathode GDL in the flow channel direction on the PEMFC performance were investigated in this study. The porosity distribution in the cathode GDL is shown in Figure 2. Case 1 was defined as a linear gradient decrease, Case 2 was defined as uniform distribution, and Case 3 was defined as a linear gradient increase. Moreover, to ensure the reliability of the comparative study, the average porosity in the three cases was kept at 0.6.

### 2.6. Boundary Conditions

The gas-counter flow was set in the channels of the cathode and anode, which was the same as the actual working conditions of the experiments. As the voltage was calculated in the constant current mode, the anode potential was set to be ground, and the cathode was set to a prescribed current density. The anode and cathode inlet were set with a constant mass flow and the outlet was set with a constant pressure outlet. The temperature of the anode and cathode current collector was set to 353.15 K, and the rest of the external boundaries were set as walls without slippage. The values of the main electrochemical parameters are listed in Table 2 and the boundary conditions are listed in Table 3.

### 2.7. Mesh Independence Study and Model Validation

Hexahedral meshes were used for all computational domains. To exclude the effect of mesh size and quantity on the calculated results, five mesh cases were considered. The voltage and calculation times were determined under the same conditions, and the results are shown in Table 4. As the number of meshes increased, the calculation results became increasingly accurate. However, the calculation time also significantly increased. To balance the accuracy and calculation time, Mesh 4 with 393,600 cells, as shown in Figure 3a, was used in this study.

To validate the reliability of the mathematical model, the polarization curves obtained through experiment and simulation were compared at an operating temperature of 353.15 K, anode/cathode relative humidities (RH) of 50% and 50%, stoichiometric ratios of 1.5 and 2.0, and pressures of 150 kPa and 150 kPa, respectively. The experimental single cell was self-assembled, composed of two graphite polar plates with single serpentine flow channels. The MEA was prepared by Wuhan Technology New Energy Co., Ltd. (Wuhan, China), its active area was 25 cm^2^, the GDL with micro-pore layer was made of Toray carbon paper, and the CCM was made of a Gore membrane, whose catalyst loads were 0.1 and 0.4 mg cm^−2^ in the anode and cathode, respectively. The porosity of the GDL was 0.6, which was the same as that of Case 2 shown in Figure 2. A Greenlight G20 test station was used in the test, which can automatically control and adjust operating conditions.

The polarization curves are shown in Figure 3b, it can be seen that the experimental data were in good agreement with the simulation results. The overall trend was similar to that of Zhang et al. [50], but the results were slightly different, due to the different working conditions. The results proved the reliability of the model.

## 3. Results and Discussions

### 3.1. Effects of Gradient Porosity Distribution on the Cell Performance

The polarization curves of the three different gradient porosity distributions in the cathode GDL are shown in Figure 4, and it can be observed that at the lower current density, the performances of three cases were almost the same. However, the difference in cell performance increased with the increase of current density. When the current density was 2400 mA cm^−2^, the cell voltage of Case 1 increased by 8% and 71% compared to Cases 2 and 3, respectively. This indicated that the GDL gradient porosity distribution mainly affected the mass overpotential at high current densities. The causes were analyzed in detail with the characteristics of the multi-physical field distribution inside the MEA.

### 3.2. Physical Field Distribution at High Current Density

The effects of the above mentioned three gradient porosity distributions in the cathode GDL on the gas–liquid transport were analyzed under the operating conditions of 2400 mA cm^−2^, T = 353.15 K, P = 251 kPa, ${\mathrm{RH}}_{\mathrm{an}}$ = 50%, ${\mathrm{RH}}_{\mathrm{ca}}$ = 50%, *ξ*_an_ = 1.5, and *ξ*_ca_ = 2.0.

#### 3.2.1. Distribution of Oxygen Molar Concentration

Figure 5a shows the oxygen molar concentration distribution of a 1/2 thickness cross-section of the cathode GDL for the three cases. The oxygen concentration decreased gradually along the air flow direction. This was because of the continuous consumption of reactants along the flow channel. The oxygen concentration under the rib was lower than the channel, owing to the higher oxygen transport resistance. Moreover, the highest oxygen molar concentration was observed in Case 1, and the lowest oxygen molar concentration was observed in Case 3. The distribution trend of oxygen concentration in CL was the same as that in GDL, but the concentration decreased, which can be seen in Figure 5b.

Figure 5c shows the streamlines in the cathode channel of Case 1. As the mass flow rate in the channel inlet for three cases was the same, the streamlines of the three cases were almost the same. Figure 5d shows the distribution of the gas velocity vector in the cathode GDL on the side of the channel inlet. It is evident that the convection in the GDL was more pronounced and the GDL gas velocity was higher in Case 1, owing to the higher porosity of the GDL on the side of the channel inlet, which had a lower transport resistance. Therefore, more oxygen entered the GDL, which resulted in a higher oxygen molar concentration in the GDL of Case 1. In Cases 2 and 3, the gas velocity in the GDL was lower, owing to the lower porosity. Hence, the oxygen concentration was lower.

To quantify the effects of GDL porosity gradient distribution on the oxygen concentration and gas flow, the GDL cross-section was divided into 10 regions of equal area along the channel direction, and the physical quantities in each region were averaged for further quantitative analysis. Figure 6a shows the oxygen concentration distribution for the three GDLs. The oxygen molar concentration in Case 1 was the highest, and in Case 3, it was the lowest. Figure 6b shows the gas velocity distribution in the GDL. The gas velocity in Case 1 was the highest, up to 5 mm·s^−1^, and that in Case 3 was the lowest, being approximately zero.

Therefore, the gradient-decreasing porosity distribution of the GDL in Case 1 increased the gas velocity in the GDL, which increased the oxygen concentration. Compared with Cases 2 and 3, the porosity distribution in Case 1 could effectively enhance the gas transport efficiency.

#### 3.2.2. Distribution of the Water Vapor Molar Concentration and Temperature

Figure 7a,b show the distribution of the water vapor molar concentration in the GDL and CL, Figure 7c is the temperature distribution in the CL. It can be seen that, on the side of the channel inlet, the vapor molar concentration under the rib was higher than that under the channel. However, the distribution at the outlet side was the opposite. This is because the water vapor in the GDL under the channel could be removed through the channel, whereas the water under the rib had a higher transport resistance and was difficult to expel on the inlet side. The airflow toward the outlet caused the water vapor in the channel to be gradually accumulated; hence, the vapor concentration under the channel was higher on the outlet side.

In addition, it was observed that the water molar concentration in the GDL and CL of Case 1 was lower than that of Case 3. On the one hand, this was because the GDL of Case 1 had a greater gas velocity along the flow channel direction, and the water produced by the electrochemical reaction was more easily expelled from the PEMFC under the purging effect of the gas flow, which resulted in the lowest water molar concentration. Therefore, the oxygen had more transport paths, which explains the higher oxygen concentration in Case 1. On the other hand, the water molar concentration is also related to temperature. Figure 7c,d show the distribution of temperature in the cross-sections at 1/2 of the cell length and on the sections at the 1/2 thickness of the cathode CL for the three cases, respectively. In the cross-sections (Figure 7c), there is only a small difference for the temperature distribution. In the longitudinal direction sections (Figure 7d), it can be seen that the temperature in the CL of Case 1 was the lowest and that Case 3 was the highest, and such a distribution is in accordance with the performance difference shown in Figure 4, because under the same current density, the cell produced more heat at a lower cell voltage. The lower the temperature, the lower the saturated vapor pressure in CL, so the lower the vapor concentration, which also meant that the CL in Case 1 had the highest oxygen concentration.

#### 3.2.3. Distribution of Liquid Saturation

Figure 8a shows the distribution of liquid saturation on both the anode and cathode sides of the MEA of Case 1. It can be seen that liquid water was mainly accumulated under the rib, owing to the higher transport resistance. Due to the oxygen concentration being highest on the side of the channel inlet, as shown in Figure 5a, the electrochemical reaction rate was higher and more water was generated. Thus, the liquid saturation on the inlet side of the cathode CL was the highest and gradually decreased along the channel direction. As the anode and cathode gases were humidified to only RH50% and were set as a counter flow pattern, because of the effect of back diffusion, the outlet area in the cathode had a lower saturation than that in the middle area in the GDL and MPL, and more liquid water accumulated in the middle. The general distribution trends of the liquid water in Cases 2 and 3 were the same, as shown in Figure 8b–d, but there were some differences of liquid saturation in the GDLs; Case 1 had the smallest value and Case 3 had the largest value (Figure 8e).

As explained above, the GDL of Case 1 had the highest gas velocity along the channel direction, which discharged the liquid water in the GDL more easily. Meanwhile, capillary pressure is also conducive to the removal of liquid water in a MEA. Figure 8f shows the capillary pressure distribution at 1/2 thickness in the cathode GDLs for the three cases. The value was the largest in the GDL of Case 1, the second largest in the GDL of Case 2, and the smallest in the GDL of Case 3, which was consistent with the saturation distribution in the GDLs of the three cases.

#### 3.2.4. Distribution of Current Density and Membrane Water

Figure 9a,b show the distribution of the current density along the channel direction for the three cases in the PEM. It can be seen that the current density in the membrane under the channel was greater than that under the rib and gradually decreased along the flow channel direction for all three cases. This was the same as the oxygen concentration distribution in MEA. The current density at the inlet side was the lowest in Case 1 and the highest in Case 3, while it had the opposite trend on the outlet side. This was because the current density was affected not only by the oxygen concentration, but also the water content in the PEM. Figure 9c,d show the distribution of the water content in the PEM, with Case 1 exhibiting the lowest and Case 3 exhibiting the highest values on the inlet side, while the opposite was observed on the outlet side. This was because, although the MEA had the highest oxygen concentration in Case 1 along the flow channel direction, the purging of the airflow at the inlet side removed more water, resulting in a lower water content, lower wettability, and higher proton transport resistance in the membrane on the inlet side. Thus, a lower current density was observed on the inlet side. At the outlet, Case 1 exhibited the highest oxygen molar concentration, and thus more water was produced, resulting in the highest water content and current density in the PEM.

It can be concluded that the GDL porosity had an important effect on the uniformity of the current density of the PEMFC. An appropriate porosity distribution can allow reasonable distribution of the reactant and change the water content distribution in the membrane, to produce a more uniform current density distribution, which can ultimately improve the durability of the PEMFC [51] Compared with Cases 2 and 3, the gradient-decreasing porosity distribution of the GDL in Case 1 reduced the water content of the membrane at the inlet side and increased the water content at the outlet side, which consequently led to lower current density differences and a higher uniformity.

### 3.3. Physical Field Distribution at Low Current Density

To comprehensively analyze the effects of the different GDL gradient porosity distributions under a lower current density, this section further investigates the different physical field distributions in the cell at a low current density of 1000 mA·cm^−2^, with no humidification of either the anode or cathode. It can be observed from Figure 10a that the GDL gas velocity along the channel direction in Case 1 was lower than that in the high current density condition, whereas the gas velocity in Cases 2 and 3 remained approximately zero. In Figure 10b, it is observed that the oxygen concentration in Case 1 was higher than that in Cases 2 and 3. However, the difference decreased, owing to less oxygen consumption at a lower current density. Figure 10c shows the distribution of liquid saturation in the GDL. It can be observed that the distribution of liquid saturation in the three GDLs at a low current density was similar to that at a high current density. Figure 10d shows the distribution of water content in the membrane. It can be seen that the water content in Case 1 was slightly lower than that in Case 3, which was related to the liquid saturation in GDLs. However, the overall membrane water content distribution of the three cases was basically the same, and the difference in the current density was small, as shown in Figure 10e.

The GDL porosity distribution had less effect on the cell performance at low current density. As having less liquid water had a lesser effect on the transport of reactants. However, more liquid water was generated and occupied the pores of the GDL at a high current density, which increased the reactant transport resistance and mass overpotential of the cell. The GDL with a gradient-decreasing porosity distribution could effectively improve the water management capability, enhancing the efficiency of the reactant transport and the performance of PEMFCs.

## 4. Conclusions

In this study, the effects of the porosity distribution of the cathode GDL, including gradient increasing, gradient decreasing, and uniformity along the channel direction, on the gas–liquid transport and performance of PEMFCs were numerically investigated. The main conclusions are as follows:

The GDL with a decreasing porosity gradient along the channel direction resulted in a GDL gas velocity of up to 0.5 cm·s^−1^ along the channel direction. The gas flow velocities in the other two GDLs with increasing and uniform gradient distributions were significantly smaller than those in Case 1. The liquid water in the GDL of Case1 could be removed more effectively owing to the greater gas velocity and capillary pressure. Therefore, it had a higher oxygen concentration, and the performance of PEMFC was improved. At a high current density (2400 mA·cm^−2^), the voltage in Case 1 increased by 8% and 71% compared to Cases 2 and 3, respectively. At a low current density, the gradient distribution of GDL porosity had less effect on the cell performance. The PEMFC with a decreasing porosity gradient in the cathode GDL along the flow channel direction, together with a anode/cathode gas counter-flow arrangement, enhanced the distribution of water content in the membrane, thus improving the uniformity of the current density distribution in the membrane along the channel direction. These conclusions not only can be used to optimize the design of a GDL but also, possibly, to enhance the durability of the PEMFC.

The effects of the gradient coordination variation of the porosity, Pt/C loading, etc., in the CL, MPL, and GDL, both along the thickness direction and flow channel direction, on the transport and the performance of the cell were much more significant, and worthy of further study in the future.

## Figures and Tables

**Figure 1 polymers-15-01629-f001:**
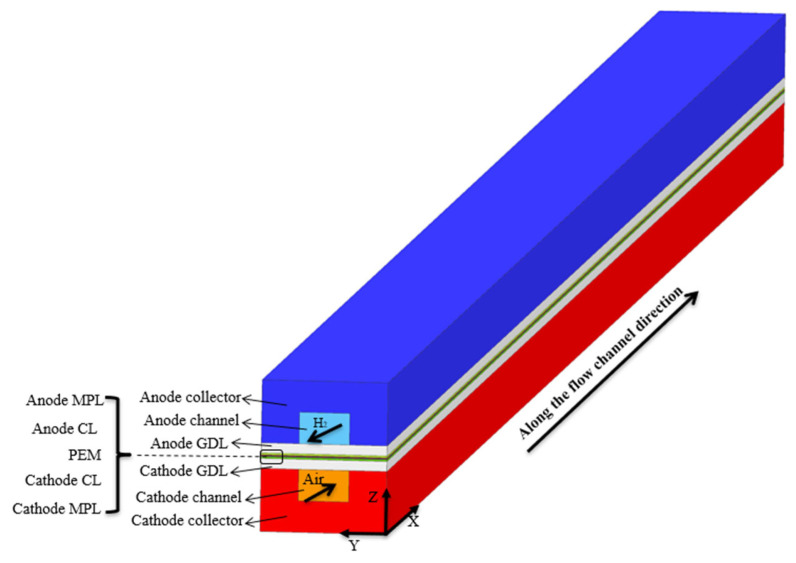
Schematic of the PEMFC model.

**Figure 2 polymers-15-01629-f002:**
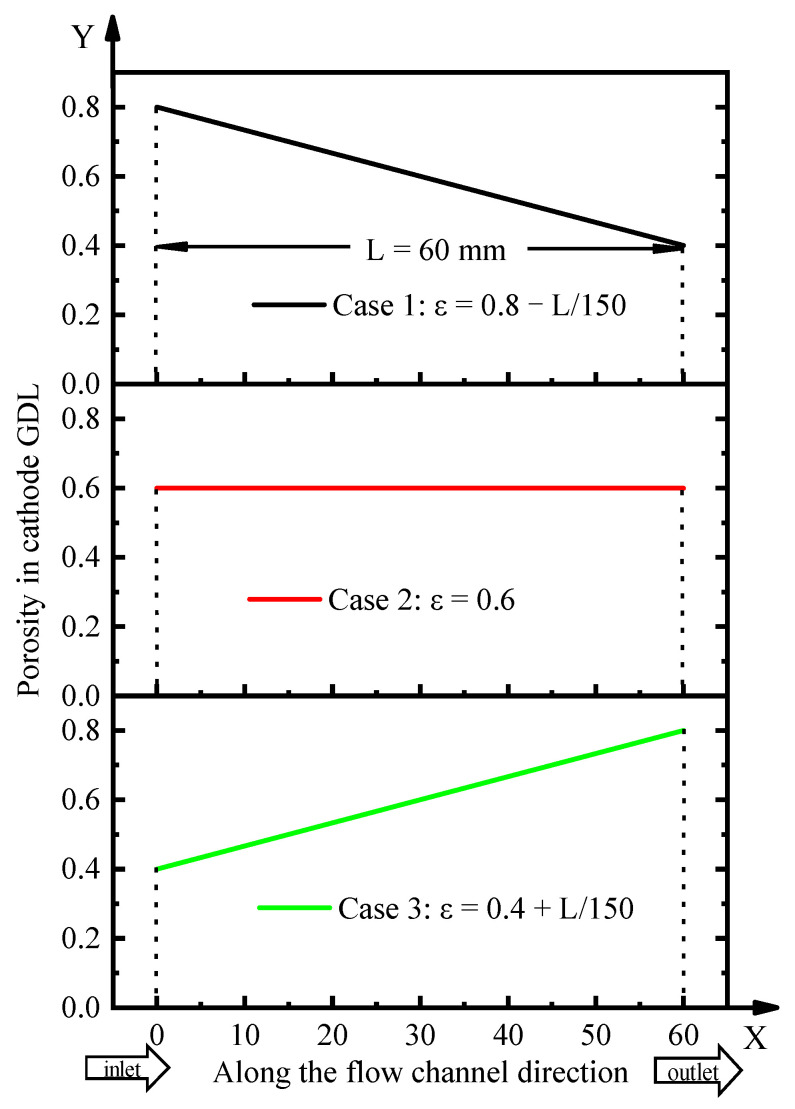
Porosity distribution of cathode GDL along the flow channel direction.

**Figure 3 polymers-15-01629-f003:**
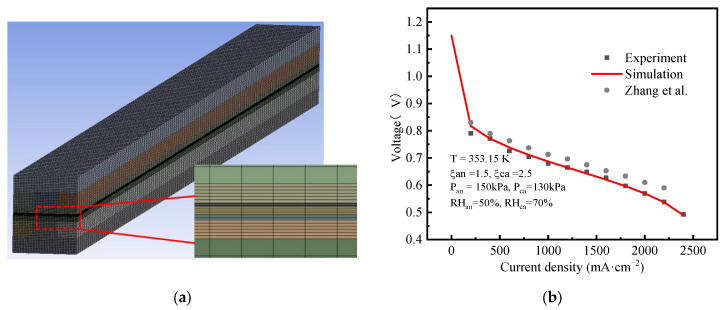
(**a**) Computational mesh used for the three cases, (**b**) Comparison of polarization curves between experiment and simulation and Zhang et al. [50].

**Figure 4 polymers-15-01629-f004:**
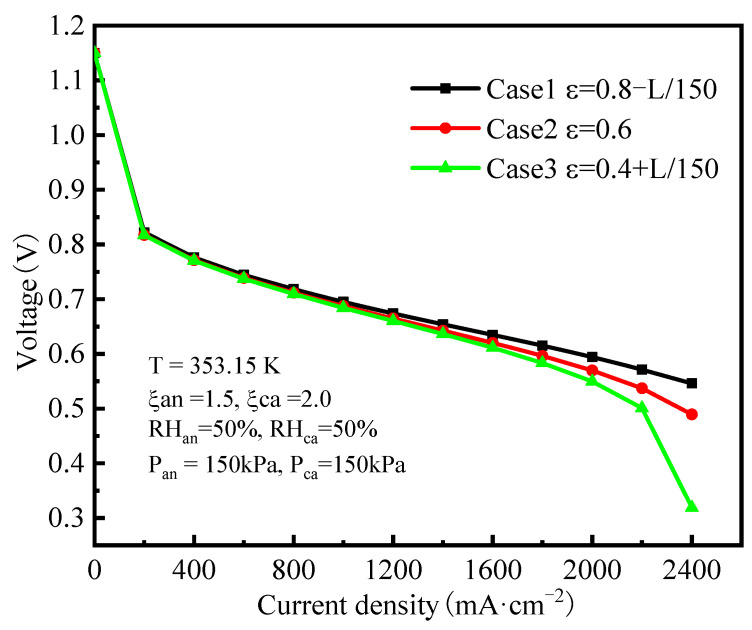
Comparison of the polarization curves.

**Figure 5 polymers-15-01629-f005:**
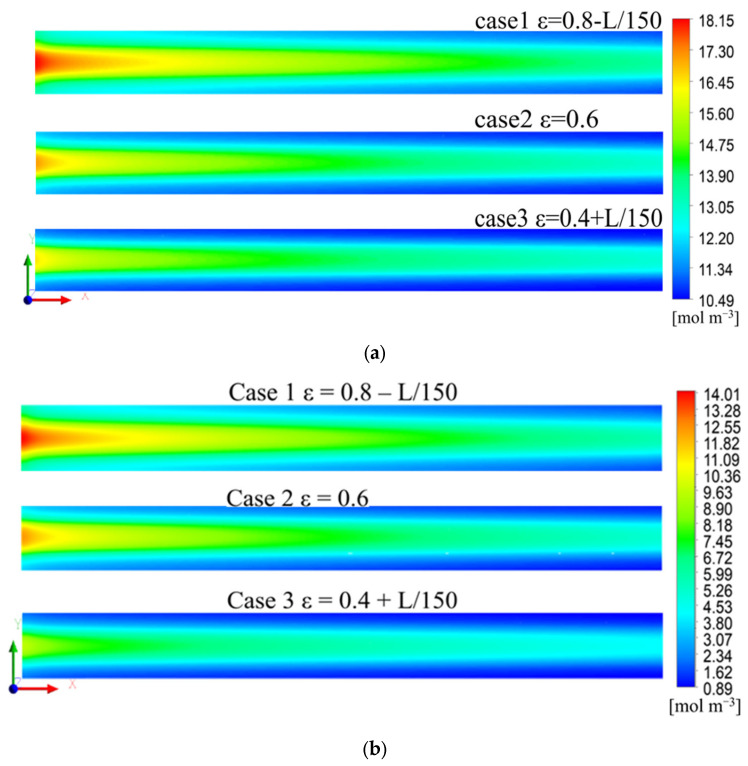
(**a**) Distribution of the oxygen concentration at the 1/2 thickness of cathode GDL, (**b**) distribution of oxygen concentration at the 1/2 thickness of cathode CL, (**c**) streamlines in the cathode channel, (**d**) distribution of gas velocity vector under the channel inlet in the cathode GDL for the three cases.

**Figure 6 polymers-15-01629-f006:**
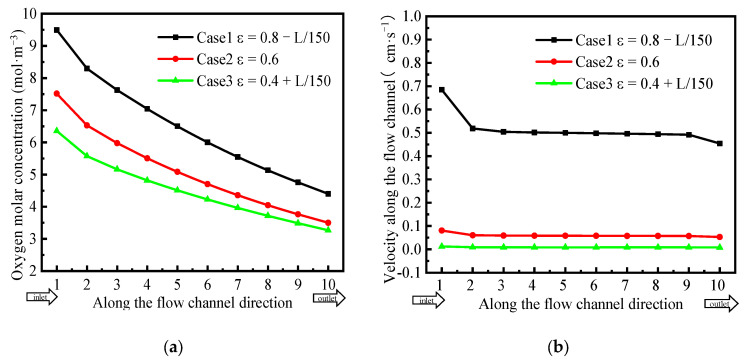
(**a**) Oxygen concentration and (**b**) gas velocity distribution at 1/2 thickness of the cathode GDL.

**Figure 7 polymers-15-01629-f007:**
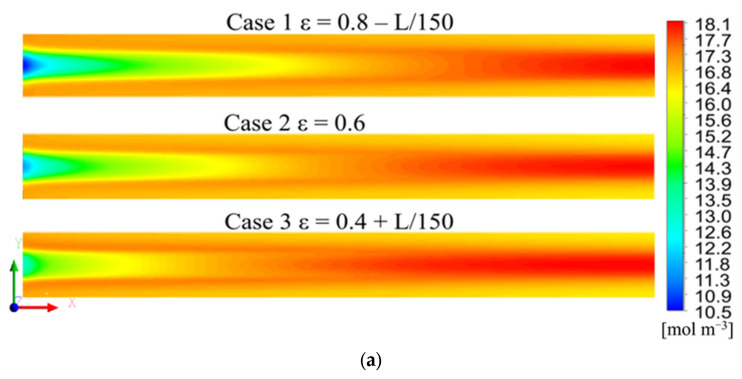
(**a**) Distribution of water vapor molar concentration at the 1/2 thickness of the cathode GDL, (**b**) Distribution of water vapor molar concentration at the 1/2 thickness of the cathode CL, (**c**) Distribution of temperature in the cross-sections at 1/2 of the cell length, (**d**) Distribution of temperature in the sections at the 1/2 thickness of the cathode CL.

**Figure 8 polymers-15-01629-f008:**
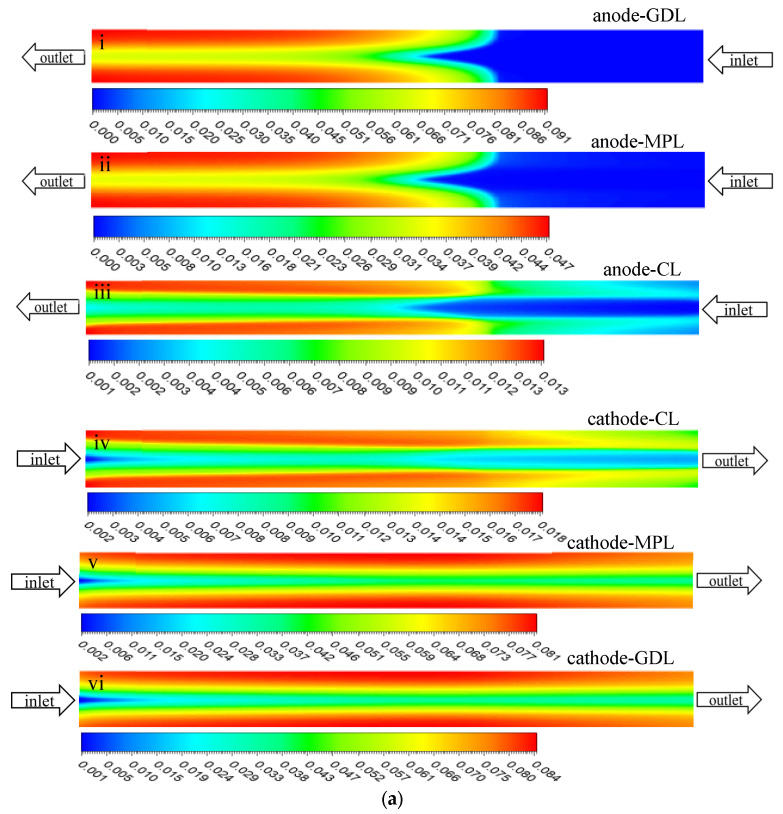
(**a**) Distribution of the liquid saturation in MEA of Case 1, (**b**–**d**) comparison of the liquid water distribution in cathode GDM for the three cases, (**e**) distribution of liquid saturation at the 1/2 thickness of the GDLs, (**f**) distribution of the capillary pressure at the 1/2 thickness of GDLs.

**Figure 9 polymers-15-01629-f009:**
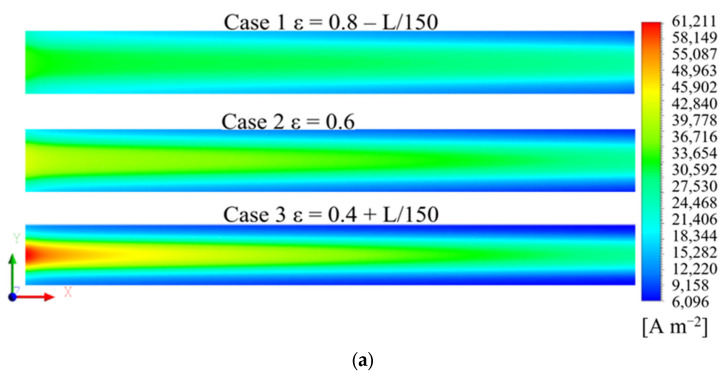
Distribution of the (**a**,**b**) current density and (**c**,**d**) water content at the 1/2 thickness of the membrane for the three cases.

**Figure 10 polymers-15-01629-f010:**
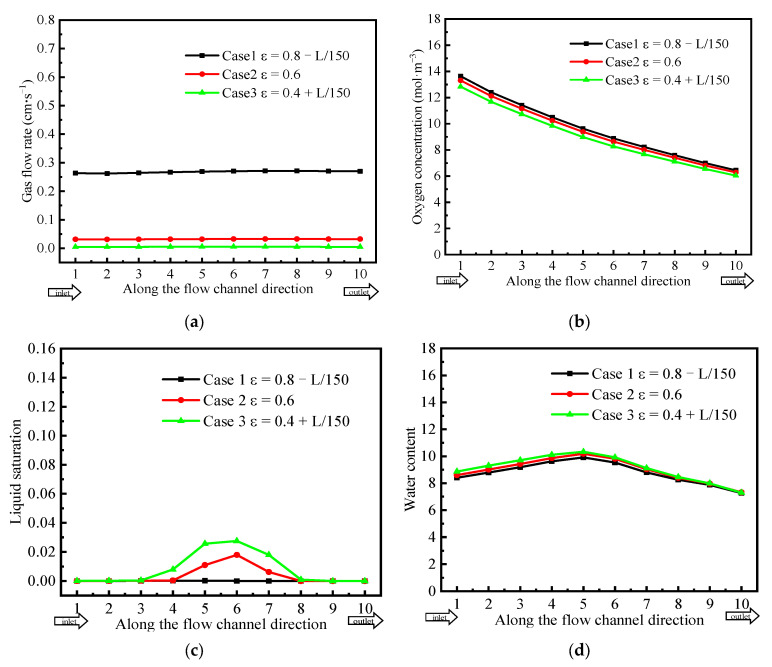
Comparison of the physical parameter distribution for the three different cases. (**a**) Gas velocity, (**b**) oxygen concentration, (**c**) liquid saturation, (**d**) water content, and (**e**) current density.

**Table 1 polymers-15-01629-t001:** Parameters of the geometric model.

Structure Parameters	Values
Length/Width/Height (mm)	60/2/2.407
Channel depth (mm)	0.5
Channel width (mm)	0.8
Rib width (mm)	0.6
GDL/MPL/CL thickness (mm)	0.16/0.03/0.009
PEM thickness (mm)	0.012
Porosity of anode GDL/MPL/CL	0.7/0.6/0.5
Porosity of cathode GDL	0.8–0.4/0.6/0.4–0.8
Porosity of cathode MPL/CL	0.6/0.5

**Table 2 polymers-15-01629-t002:** Physical and electrochemical parameters [48,49].

Parameters	Value
GDL/MPL/CL Permeability (m^2^)	9.1 × 10^−12^/3.4 × 10^−12^/3 × 10^−14^
GDL/MPL/CL electrical conductivity (s·m^−1^)	1826.7/3779.2/1162.5
GDL/MPL/CL thermal conductivity (W·m^−1^·K)	2.6/5.4/9.3
GDL/MPL/CL specific heat capacity (J·kg^−1^·K)	710/710/3300
GDL/MPL/CL contact angle (deg)	130/140/120
GDL/MPL/CL density (kg·m^−3^)	440/1880/1000
Active area (mm^2^)	120
Reference anode exchange current density (mA·cm^−2^)	1 × 10^5^
Reference cathode exchange current density (mA·cm^−2^)	100
Surface/Volume ratio in CL (m^−1^)	2 × 10^−5^
Volume fraction of the ionomer in the CL	ω = 0.25
Anode transfer coefficient $\alpha_{an}$	0.7
Cathode transfer coefficient $\alpha_{ca}$	0.3
Effective electron conductivity and ion conductivity	$\kappa_{\mathrm{ion}}^{\mathrm{eff}}=\omega^{1.5}\kappa_{\mathrm{ion}};\;\kappa_{\mathrm{ele}}^{\mathrm{eff}}={(1-\varepsilon -\omega )}^{1.5}\kappa_{\mathrm{ele}}$
Electro-osmotic drag (EOD) drag coefficient	$n_{d}=\frac{2.5\lambda }{22}$
Reference hydrogen and oxygen concentrations	$C_{O_{2}}^{\mathrm{ref}}=C_{H_{2}}^{\mathrm{ref}}={40\mathrm{mol}\;m}^{-3}$
Gas dynamic viscosity (${\mathrm{kg}\;m}^{-1}s^{-1}$)	$\mu_{g}^{a}=1{.53\;\times \;10}^{-5};$ $\;\mu_{g}^{c}=1{.79\;\times \;10}^{-5}$
Liquid water dynamic viscosity (mPa·s) (at 353.15 K)	$\mu_{l}$ = 0.356

**Table 3 polymers-15-01629-t003:** Boundary conditions.

Parameters	Value
H_2_ mass fraction in the anode inlet	$W_{H_{2}}=M_{H_{2}}\left[1-p_{sat}\cdot RH_{an}/P\right]$
O_2_ mass fraction in the cathode inlet	$W_{O_{2}}=0.21\cdot M_{O_{2}}\left[1-p_{sat}\cdot RH_{ca}/P\right]$
Mass flow in the anode inlet (kg·s^−1^)	$m_{an}=\xi_{an}\frac{i\cdot A\cdot M_{H_{2}}}{2F\cdot W_{H_{2}}}$
Mass flow in the cathode inlet (kg·s^−1^)	$m_{ca}=\xi_{ca}\frac{i\cdot A\cdot M_{O_{2}}}{4F\cdot W_{O_{2}}}$
Relative humidity in anode/cathode	50%/50%
Outlet pressure in anode/cathode (kPa)	150/150
Stoichiometry in anode/cathode	1.5/2.0
Temperature (K)	353.15

**Table 4 polymers-15-01629-t004:** Results of mesh independence study at 2400 mA cm^−2^.

	Number of Mesh	Voltage (V)	Calculation Time (h)
Mesh 1	132,000	0.517574	2.7
Mesh 2	264,000	0.520874	5.2
Mesh 3	393,600	0.523761	6.1
Mesh 4	787,200	0.524113	9.2
Mesh 5	924,000	0.524419	13.6

## Data Availability

No new data were created.

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
