# Peer review of "Effects of Cathode GDL Gradient Porosity Distribution along the Flow Channel Direction on Gas–Liquid Transport and Performance of PEMFC"

_polymers, 2023, doi:10.3390/polym15071629_

Round 1

Reviewer 1 Report

This is a very sound paper on transport phenomena modeling. It is a very educational and enjoyable read, in which the authors are able to explain very clearly and thoroughly the transport models, the equations, and the assumptions. This paper can be accepted as is. HOWEVER, I would suggest that this paper to be moved and be published in chemical engineering education types of journal, such as ChemEngineering in mdpi, where it would be able to reach broader audience and to be more impactful.

I would also suggest that the authors would change the title of this manuscript, so that chemical engineering students who are looking for transport phenomena paper examples can easily find this paper.

Author Response

Point 1: I would also suggest that the authors would change the title of this manuscript, so that chemical engineering students who are looking for transport phenomena paper examples can easily find this paper.

Response 1: Thank you very much for your suggestion. We thought over the title. PEMFC covers chemical engineering, engineering thermo-physics and other areas. According to the main contents studied in this paper, we think the title is suitable. We would like to change the title of this manuscript if the reviewer can give us more detailed guidance.

Reviewer 2 Report

The authors presented a numerical study on the Effects of cathode GDL gradient porosity distribution along the flow channel direction on gas–liquid transport and performance of PEMFC.

The novelty of the paper is to be clearly stated.

The boundary conditions are to be expressed mathematically.

More details about the numerical method are to be provided.

A figure presenting the 3D mesh is to be added.

A qualitative (flow structure for example) verification/validation of the numerical model is to be performed by comparing with earlier published results.

The 3D flow structure (streamlines) is to be plotted and described.

What is the considered range for Reynolds number.

In Figure 1, correct ‘’anode’’ instead of ‘’abode’’

Results related to the temperature profile are to be presented on the whole domain and not the fluid domain.

The neglection of the effect of the gravity is to be justified.

The paper is to be checked against misprints and grammatical mistakes.

Author Response

Point 1: The novelty of the paper is to be clearly stated.

Response 1: Thank you very much for your suggestion. We have further stated the novelty of this paper, which have been marked in red and bold in the abstract.

Point 2: The boundary conditions are to be expressed mathematically.

Response 2: Thanks for your kindly suggestion, we have made corresponding corrections about the boundary conditions which have been expressed mathematically in the Table 3.

Point 3:
More details about the numerical method are to be provided.

Response 3: We appreciate your suggestion very much, and we have made a more detailed description of the calculation method of the numerical model. The additional expressions are marked in red and bold in section 2.4 of the revised version manuscript.

Point 4: A figure presenting the 3D mesh is to be added.

Response 4: Thanks for your kindly suggestion, the figure of 3D mesh have been replaced the previous figure according to your suggestion. It can be seen in the Figure 3(a).

Point 5: A qualitative (flow structure for example) verification/validation of the numerical model is to be performed by comparing with earlier published results.

Response 5: Thanks for your helpful suggestion. We compared the results of model validation with those of Zhang et al [53], thus increasing the reliability of the results.

[53] Zhang H, Xiao L, Chuang P-YA, Djilali N, Sui P-C. Coupled stress–strain and transport in proton exchange membrane fuel cell with metallic bipolar plates. Applied Energy. 2019;251.

Point 6: The 3D flow structure (streamlines) is to be plotted and described.

Response 6: Thanks for your helpful suggestion. We have plotted the 3D flow structure (streamlines) in the Figure 5 (c). Due to the same mass flow rate in the cathode channel inlet for three cases, the difference is small.

Point 7: What is the considered range for Reynolds number.

Response 7: Thanks for your kindly suggestion. The fluid can be considered as the laminar flow when the Reynolds number is less than 2000. Since the maximum Reynolds number of the fluid in this paper is 47, much less than 2000, it is considered as laminar flow.

Point 8: In Figure 1, correct ‘’anode’’ instead of ‘’abode’’

Response 8: We appreciate very much for your correction. The wrong expression has been corrected in the Figure 1.

Point 9: Results related to the temperature profile are to be presented on the whole domain and not the fluid domain.

Response 9: Thanks for your useful suggestion. We have conducted an in-depth analysis of your question. The operating temperature of the cell is 80 ℃, and the temperature in the terminal face of the anode and cathode is set as constant (80 ℃). We have plotted the temperature distribution on the cross sections at 1/2 of the cell length and 1/2 thickness of the cathode CL. The temperature difference is mainly on the CLs. which can be seen in the Figure 7(c).

Point 10: The neglection of the effect of the gravity is to be justified.

Response 10: Thanks for your good question, the effect of the gravity is complex, which is related to the size of the channel, the gas velocity, the wall hydrophilicity, etc. For simplicity, the effect of gravity is ignored in our study, just as [34,44] done.

[34] Rostami L, Haghshenasfard M, Sadeghi M, Zhiani M. A 3D CFD model of novel flow channel designs based on the serpentine and the parallel design for performance enhancement of PEMFC. Energy. 2022;258.

[44] ANSYS Fluent 2022 R1 PEMFC Fluent help.

Point 11: The paper is to be checked against misprints and grammatical mistakes.

Response 11: Thank you for your reminder. We invited a English professor to help us and have double checked the whole manuscript.
